Does reef crest zone selection influence Acropora palmata (Lamarck, 1816) fragment survival and growth?

Ramos Romero Amanda 1 2
González-Díaz Patricia 1 3
Aguilera Pérez Gabriela 1
Banaszak Anastazia T. banaszak@cmarl.unam.mx 2
1 Centro de Investigaciones Marinas, Universidad de La Habana , La Habana , Cuba
2 Unidad Académica de Sistemas Arrecifales, Universidad Nacional Autónoma de México , Puerto Morelos , Quintana Roo , Mexico
3 Harte Research Institute for Gulf of Mexico Studies, Texas A&M University , Corpus Christi , TX , United States of America
Berges John
Electronic publication date: 2025 Nov 14
Publication date: 2025
Volume: 13
Electronic Location ID: e20303
Received 2025 Apr 7; Accepted 2025 Oct 7
Copyright: ©2025 Ramos Romero et al.
Copyright year: 2025
Copyright holder: Ramos Romero et al.
License: This is an open access article distributed under the terms of the Creative Commons Attribution License, which permits unrestricted use, distribution, reproduction and adaptation in any medium and for any purpose provided that it is properly attributed. For attribution, the original author(s), title, publication source (PeerJ) and either DOI or URL of the article must be cited.
License URL: https://creativecommons.org/licenses/by/4.0/

Keywords: Acropora, Crest zones, Field experiment, Fragments, Restoration, Water flow

Funding: Consejo Nacional de Ciencia y Tecnología, México Centro de Investigaciones Marinas Universidad de La Habana Ocean for Youth (OFY) Sweet-Avalon Harte Research Institute at Texas A & M University-Corpus Christi The Ocean Foundation, the Environmental Defense Fund (EDF) Posgrado en Ciencias del Mar y Limnología, UNAM, Mexico Amanda Ramos Romero was supported by a doctoral scholarship from the Consejo Nacional de Ciencia y Tecnología, México. Financial and logistical support were provided by Centro de Investigaciones Marinas, Universidad de La Habana, Ocean for Youth (OFY), and Sweet-Avalon. Funding was provided by the Harte Research Institute at Texas A & M University-Corpus Christi, The Ocean Foundation, the Environmental Defense Fund (EDF), and Posgrado en Ciencias del Mar y Limnología, UNAM, Mexico. The funders had no role in study design, data collection and analysis, decision to publish, or preparation of the manuscript.

==============================
In this study, we evaluated the effects of the crest zones on the survival and growth of Acropora palmata fragments in four sites differentially impacted by multiple natural and anthropogenic stressors. The crests are in the northwest (Playa Baracoa and Rincón de Guanabo) and the south-central region (El Peruano and Mariflores in Jardines de la Reina National Park) of Cuba. We established a field-based experiment with 50 fragments placed in each crest, 25 fragments in the back crest zone and 25 in the fore crest zone, parallel to the shoreline. The water flow intensity was estimated in both crest zones, using the dissolution of plaster discs as an indicator. The survival and growth of fragments were significantly influenced by site-specific and microhabitat conditions. The survival of the A. palmata fragments was high (survival probability: >0.6) in all four crests. Fragments placed in the fore crest zone (p = 0.02) exhibited higher survival than those in the back crest zone. The growth rates were slower (−1.5 to 7.3 cm year−1) than those previously recorded for wild A. palmata colonies and were negatively affected (estimate = −6.1; p = 0.004) in the fore crest zone. The dissolution of the plaster discs did not indicate a marked gradient of water flow between the crest zones, during April and June, but it was significantly higher (p = 0.03) in the fore crest zone in December at El Peruano and Mariflores crests, indicating temporal variations during the monitoring periods. The dissolution of plaster discs, as a proxy for water flow, did not have a significant effect on fragment survival between zones. However, when the dissolution was below approximately 68%, growth declined slightly; above this threshold, growth increased, possibly by higher inferred water flow. At higher levels of water flow (dissolution of the plaster discs), the positive effect of temperature on growth was attenuated (p = 0.007), suggesting that under strong water flow heat may be dissipated. These results highlight the importance of conducting small-scale pilot studies to identify the microhabitat conditions and to select effective restoration sites. We recommend that future restoration efforts should integrate local ecological knowledge with in situ environmental measurements to enhance coral fragment survival and growth, and to improve the long-term success of restoration interventions under variable and changing reef conditions.

Introduction

The reef-building coral Acropora palmata (Lamarck, 1816) is listed as Critically Endangered by the International Union for Conservation of Nature (IUCN) since 2008 (Aronson et al., 2008). Currently, A. palmata populations are considered not resilient enough to recover naturally due to the impact of various anthropogenic and natural factors such as water pollution, fishing pressure, coral diseases and increases in ocean temperature (Jackson et al., 2014; Dutra et al., 2021). As a consequence, restoration programs have been implemented as a strategy to recover Acropora spp. populations throughout the Caribbean (Ladd et al., 2018; Boström-Einarsson et al., 2020).

Coral reef restoration initiatives aim to halt the decline in coral cover, and recover the structural complexity of reefs, ecological functions and ecosystem services, thereby supporting its resilience (Lirman & Schopmeyer, 2016; United Nations Environment Assembly UNEA , 2019). However, when a system has undergone shifts beyond its natural state, restoring it to its original condition may be impractical or even unfeasible (Rinkevich, 2014). In such cases, restoration efforts should prioritize the development of alternative systems that fulfill desired ecological attributes while sustaining the goods and services formerly provided by the original system (Jackson & Hobbs, 2009).

Most coral restoration studies have focused on evaluating the survival and growth of outplanted coral fragments. Outplanted corals generally exhibit high survival rates (66%), with Acropora spp. reaching up to 90%. Most restoration projects (60%) last from 12 to 18 months and an average area of approximately 100 m2 (Young, Schopmeyer & Lirman, 2012; Bayraktarov et al., 2020). These findings suggest that most restoration initiatives are implemented at relatively small temporal and spatial scales. Nevertheless, successful long-term efforts, extending up to three or even 12 years, have shown increases in coral cover, and fish diversity and abundance (Bayraktarov et al., 2020; Boström-Einarsson et al., 2020).

Despite these efforts, restoration practitioners have prioritized maximizing the number of outplanted corals, while overlooking the implementation of experimental designs that could support restoration to optimize reef recovery (Ladd, Burkepile & Shantz, 2019). A disconnect remains between the ecological knowledge generated and its practical application in the selection of restoration sites (Miller, 2002; Shaver & Silliman, 2017). Site selection is often based on logistical or qualitative criteria, without considering measurable environmental and ecological variables such as temperature, input of nutrients, water flow, habitat structure, sedimentation, benthic composition, and herbivory or corallivory processes, despite their potential influence on the performance of outplanted fragments (Hein et al., 2017; Ladd et al., 2018).

Improving restoration practices for threatened coral species requires identifying the environmental conditions that influence coral growth and mortality. Furthermore, quantifying the magnitude of these effects is critical for informing targeted management and conservation strategies (Enochs et al., 2014). Among these drivers, wave energy constitutes a fundamental physical force determining reef function and ecology (Madin & Connolly, 2006; Simonson et al., 2021). Wave breaking increases the water level, establishing a pressure gradient that drives flow across the reef (Lowe et al., 2009). In reefs, the wave dissipation can be high (86% in crests, Ferrario et al., 2014) due to the complexity of the substrate (Monismith et al., 2015; Rogers et al., 2016). Reef crests exposed to wave action retain less pollutants, have higher recruitment rates, and exhibit faster coral growth (Sebens, 1991). Waves increase water movement, influencing calcification, morphology, photosynthesis, respiration, and particle capture by the corals (Dennison & Barnes, 1988; Patterson, Sebens & Olson, 1991; Lesser et al., 1994).

The coral A. palmata is adapted to high energy environments (Done, 1982; D’Antonio, Gilliam & Walker, 2016) such as reef crests. The direction and distribution of its branches, morphology, and growth vary depending on waves and water movement. These adaptive strategies of the A. palmata colonies minimize the damage caused by wave force and improve coral survival (Graus, Chamberlain & Boker, 1977; Precht & Miller, 2007). Our aim was to determine if outplanting Acropora palmata fragments in two zones of the reef crest influenced their growth and survival.

Materials and Methods

Study area

The study was carried out in four shallow reef crests in Cuba. Two of them are in the northwest region, in Playa Baracoa, Artemisa Province (23°03′20″N, 82°33′10″W) and Rincón de Guanabo, La Habana Province (23°10′23.63″N, 82°05′57.46″W). The other two, Mariflores (20°46′17.46″N, 78°53′44.34″W) and El Peruano (20°50′46.74″N, 79°1′4.32″W) lie to the south of the central region of Cuba, in Jardines de la Reina National Park (Fig. 1). These crests differ in distance from shore, abundance, and diversity of species such as fish and sea urchins, anthropogenic stressors, management and protection (Pina-Amargós et al., 2014; Rey-Villiers, Sánchez & González-Díaz, 2021; Ramos et al., 2024).

Figure 1 Location of crests where the experiments were undertaken.

The map of Cuba shows Playa Baracoa (circle), Rincón de Guanabo (square), Mariflores (star) and El Peruano (triangle) crests. Map data©2024 Google Earth, Image Landsat/Copernicus, Data SIO, NOAA, US Navy, NGA, GEBCO; Image©2024 Airbus; Image©2024 Maxar Technologies.

The reefs in the northwestern region are impacted by their proximity to the capital city and to coastal development (González-Díaz et al., 2018; Ramos et al., 2024). Pollution from heavy metals and fertilizers, street runoff, the Almendares and Quibú rivers and Havana Bay are the main anthropogenic factors that affect reefs near the city (González-Díaz, De la Guardia & González-Sansón, 2003; Rey-Villiers et al., 2020; Rey-Villiers, Sánchez & González-Díaz, 2021; Ramos et al., 2024). The Playa Baracoa crest is located approximately 230 m from the shoreline and 2 km east of Santa Ana River where untreated wastewater from a local educational institution (Latin American School of Medicine with an average annual enrollment of 10,000 students) is released (Ramos et al., 2024). Rincón de Guanabo is a marine protected area (Protected Natural Landscape/seascape similar to category V IUCN) located 800 m from the coastline. However, there is no effective management plan for this crest. The crest is nearly three km east of an oil drilling and extraction area (Boca de Jaruco thermoelectric power station), but data on nutrient load or pollutants (e.g., hydrocarbons) are either absent or unavailable. In these crests, coral cover is low (<17%) and algal cover is high (>87%) (Ramos et al., 2024). Fish biomass is low (∼12 g m−2), due to subsistence fishing (Duran et al., 2018; Gil-Agudelo et al., 2020).

On the other hand, Mariflores and El Peruano crests are located approximately 80 km from the main island of Cuba in the Jardines de la Reina National Park. This National Park has low human impact and is in good condition in terms of conservation (Beyer et al., 2018). It has been classified as an oligotrophic system where nutrient input is due to organic matter from mangroves, muddy sediments, and open waters such as the Gulf of Ana María and the Caribbean Sea (Pina-Amargós, Figueredo-Martín & Ross, 2021). In Jardines de la Reina National Park, the reef crests are characterized by A. palmata cover ranging from 22% to 45%, algal cover between 22% and 49%, and Diadema antillarum densities of 0.3 to 4.7 ind m−2 (Hernández-Fernández, López & Sotolongo, 2016). Fish abundance is high due to the absence of overfishing (Pina-Amargós, Figueredo-Martín & Ross, 2021). According to Pina-Amargós et al. (2014), the crests showed high abundances of commercially important fish species, including Epinephelus striatus (15–65 cm in length, 0.2 ± 0.02 ind 1,000 m−2), Lutjanus cyanopterus (25–85 cm, 0.2 ±0.03 ind 1,000 m−2), and L. apodus (10–55 cm, 53.2 ± 2.2 ind 1,000 m−2). Large herbivorous fish were also present, such as Scarus guacamaia (45–115 cm, 0.1 ± 0.02 ind 1,000 m−2) and Sc. coelestinus (39–105 cm, 0.08 ± 0.003 ind 1,000 m−2).

Experimental design

Along the crest, we identified two zones relative to the breaking wave and parallel to the shoreline: the fore zone, where wave breaks first, and the back zone, the opposite side of the crest, where the energy of the wave is dissipated by the fore zone (Fig. 2). To estimate water flow intensity in the fore and back crest zones we used the dissolution of plaster discs (or ‘clod cards’ Doty, 1971) as an indicator (Jokiel & Morrissey, 1993). On each crest, we placed eight plaster discs weighing approximately 144 g each. Four plaster discs were placed parallel to the shoreline in the fore crest zone and another four were placed in the back crest zone. The discs were mounted on steel rods 10 cm off the substrate using a wire. The plaster discs were kept for 48 h in Playa Baracoa and Rincón de Guanabo crests, during April and June 2023, respectively. Whereas in El Peruano and Mariflores crests, the discs were placed twice, in December 2022 and April 2023, for 48 h. An exception was in December in Mariflores when the discs were retrieved after 24 h for logistical reasons. Once the discs were removed, they were left to dry and reweighed. The weight lost from each plaster disc provided an indicator of energy on each crest zone.

Figure 2 Drone image of (A) Mariflores and (B) El Peruano crests which lie to the south of the central region of Cuba, in Jardines de la Reina National Park.

The yellow crosses represent the Acropora palmata fragments placed in the fore and back crest zones and the red dashed arrows show the direction of waves breaking on each crest. Photo credit: Noel Lopez Fernandez.

To test the effect of the crest zones associated with the water flow gradient on the survival and growth of A. palmata fragments, we established a field experiment. The fragments were collected from randomly selected colonies within each study crest. They were cut with forceps from the apical portions of different branches of several colonies, exhibiting a roughly rectangular shape and lacking secondary branching. These fragments were subsequently outplanted onto the same reef crest as the parental colonies, rather than in a common garden. A total of 50 fragments with a size (width and height) between one and seven cm were placed in each crest, 25 fragments in the back and 25 in the fore crest zones, one m apart each and parallel to the shoreline (Fig. 2). The A. palmata fragments were attached to the substrate with epoxy (KLIPTON ACUAPLAST) and each was labeled and monitored over time. The depth at which the A. palmata fragments were placed varied among crests and between zones within each crest. In Playa Baracoa, the depth is approximately 1.5 m and in Rincón de Guanabo, around 2 m, in both zones. In El Peruano, the depth is one m in the back crest zone and 2.5 m in the fore crest zone, while in Mariflores, the back crest zone has a depth of approximately one m and the fore crest zone about 1.8 m.

In Playa Baracoa, the experiment with A. palmata fragments was carried out in January 2022. The maximum heights and widths of the fragments were measured with a Vernier caliper at 0, 152, 279, and 453 days. In Rincón de Guanabo, the experiment began in January 2022, and the fragments were monitored at 0, 136, 261, and 433 days. In El Peruano, the experiment started in February 2022 and fragments were measured at 0, 172, 291, and 423 days. In Mariflores, the experiment began in February 2022, and the fragments were measured at 0, 168, 291, and 423 days. The number of fragments measured varied in each period, because some died, were detached from the substrate or were not found on the reef (due to human error) but were relocated in the following monitoring period.

The temperature was recorded every 30 min during the monitoring period using a data logger (HOBO UA-002-64 Pendant) deployed at Playa Baracoa and Rincón de Guanabo crests. In Jardines de la Reina, data were obtained from a logger located on a crest between El Peruano and Mariflores, as it was not possible to deploy loggers on each crest. The data logger records do not precisely align with the coral outplanting start date, either due to the unavailability of loggers at that time or because some units failed, resulting in the loss of temperature data. The mean monthly temperature was determined considering all records obtained every 30 min during each month.

Data analysis

Statistical analyses were conducted using the R program, created by the R Core Team (2016, version 4.0.5). The non-parametric Wilcoxon-Mann–Whitney test was used to test differences in the dissolution of the plaster discs between the crest zones. The median dissolution percentage was calculated from the four plaster discs in each zone. The Student t test (parametric) or the Wilcoxon-Mann–Whitney test (non-parametric) were used to determine differences in the initial size of the fragments between the back and fore crest zones. The Kruskal–Wallis test (non-parametric) and the Dunn post hoc test were used to determine differences in the initial sizes of the fragments between the four crests. A Cox regression model, coxph(Surv(survival time, survival) ∼ height + width, data = Data) was used to analyze whether the initial size of the fragments influenced survival, using the survival package (Therneau & Grambsch, 2000). The survival probability (sp) of the fragments was evaluated using the nonparametric Kaplan–Meier estimator with the survminer package (Kassambara, Kosinski & Biecek, 2021). This estimator models survival probability over time by accounting for the exact timing of mortality events. It partitions the follow-up period into intervals defined by the occurrence of events (e.g., fragment mortality) and estimates the conditional survival probability for each interval. These conditional probabilities are then multiplied sequentially to obtain the cumulative survival function. This approach incorporates both time-to-event information and censored data.

The growth rate of the fragments was determined for both width and height during each study period, using the formula of Mercado-Molina, Ruiz-Diaz & Sabat (2014): Growth rate=initial size−final sizetime

where:

initial size: is the height and/or width(cm) of fragment at the beginning of each period.

final size: is the height and/or width (cm) reached by fragments at the end of each period.

time: is the number of days included in each study period.

We tested for homogeneity of variance and normality using Levene’s and Shapiro–Wilk’s tests, respectively (R package nortest). The growth rate data did not fit a normal distribution, and we performed a square root transformation. The t-Student (parametric) test was used to determine differences in the growth rates between the back and fore crest zones. A generalized and linear mixed model was used to estimate the effect of: (1) the water flow represented by the dissolution of the plaster discs, (2) mean temperature in each period, (3) depth, (4) period, (5) crest zone and (6) crest site on fragment survival and growth rate, respectively with the lme4 package (Bates et al., 2015). For this model the reference level for the crest site and zone factors were Mariflores crest and the back zone respectively, with an estimate value of 9.4. A segmented model was used to identify dissolution values from which the effect on growth begins to change using the segmented package (Muggeo, 2008).

Results

Dissolution of the plaster discs

The dissolution of the plaster discs was similar between the fore and back crest zones from the northwestern and the central regions of Cuba during April and June. In El Peruano and Mariflores reefs, the dissolution of the plaster discs varied significantly (p = 0.03) between crest zones, during December. In El Peruano, the dissolution was 38% and 49% in the fore and back crest zones, respectively. In Mariflores, the dissolution of the plaster discs was lower than in El Peruano, at 35% in the fore crest zone and 28% in the back crest zone (Table 1).

Table 1 Median dissolution percentages (%) and range (minimum, maximum) of the plaster discs located in the fore and back zones of the crests (PB, Playa Baracoa; RG, Rincón de Guanabo; Pr, El Peruano and Mf, Mariflores).

The p value ≤ 0.05 indicates significant differences between the crest zones, indicated in bold.

			Dissolution percentage (%) & range (min, max)		
Crest	Month	Time (hrs)	Back zone	Fore zone	p-value	
PB	April	48	63 (61, 69)	68 (57, 72)	0.6	
RG	June	48	64 (61, 68)	60 (48, 62)	0.1	
Pr	December	48	38 (32, 43)	49 (41, 50)	0.03	
	April	48	89 (81, 93)	95 (85, 98)	0.3	
Mf	December	24	28 (20, 29)	35 (29, 39)	0.03	
	April	48	65 (61, 74)	67 (66, 74)	0.5	

Initial size of the fragments

The initial mean width and height of the fragments ranged from two to four cm and was similar between the fore and back crest zones of the reefs, except in Mariflores where the initial width of the fragments varied significantly (p = 0.003) between zones (Table S1). The initial size varied significantly across the four reefs (Table S2). The Cox regression model did not show an effect of the initial size of the fragments on the survival of fragments in the four reefs.

Survival

Overall, the survival probability of fragments was high (sp > 0.6) in the four crests. The survival was similar among the four fore crest zones ranging from 0.9 to 0.6. However, survival varied significantly (p = 0.03) among the back crest zones. In this case, Rincón de Guanabo had the lowest (sp = 0.3) survival at 347 days in the back crest zone (Table 2, Fig. 3). The generalized linear mixed model did not show a significant effect of dissolution of the plaster discs, mean temperature, depth, or period on fragment survival. However, the survival was positively influenced by the four fore zones (estimate = 5.3; p = 0.02), and Playa Baracoa (estimate = 6.2; p = 0.002) and El Peruano (estimate = 12.9; p = 0.0002) crests, and the interaction between the fore crest zone and the Rincón de Guanabo crest (estimate = 6.9; p = 0.03).

Table 2 The survival probability (sp) of Acropora palmata fragments for the back and fore crest zones during the study periods in Playa Baracoa (PB), Rincón de Guanabo (RG), El Peruano (Pr) and Mariflores (Mf) reefs.

The p value ≤ 0.05 indicates significant differences between the crest zones, indicated in bold.

		Survival probability		
Reef	Time (days)	Back	Fore	p-value	
PB	76	0.9	0.9	0.3	
	215	0.8	0.9		
	366	0.7	0.8		
	453	0.6	0.8		
RG	68	0.6	1	0.004	
	199	0.4	0.8		
	347	0.3	0.7		
Pr	24	0.9	1	0.9	
	110	0.8	1		
	232	0.7	0.6		
Mf	23	0.8	0.9	0.1	
	107	0.7	0.9		
	232	0.7	0.9		

Figure 3 Kaplan–Meier estimated survival probabilities for fore (red) and back (blue) crest zones in Playa Baracoa (A), Rincón de Guanabo (B), Mariflores (C), and El Peruano (D) reefs.

The p-value ≤ 0.05 indicates significant differences between crest zones.

Growth rates

The linear model showed that the growth rates were negatively affected (estimate = −6.1; p = 0.004) in the fore crest zone. The rate of increase in the height of the fragments was four-fold higher in the back crest zone in Playa Baracoa (back: 2.9 ± 2.9 cm year−1, fore: 0.7 ± 2.2 cm year−1, p = 0.03) and five-fold greater in Rincón de Guanabo (back: 3.7 ± 2.2 cm year−1, fore: 0.7 ± 2.6 cm year−1, p = 0.02) relative to the fore crest zone during the third period. In El Peruano, the rate of increase in the width of the fragments was six-fold greater in the back crest zone (2.2 ± 1.8 cm year−1) with respect to the fore crest zone (0.4 ± 2.2 cm year−1). Similarly, in Mariflores, the rate of increase in the width of the fragments of the back crest zone (3.3 ± 2.6 cm year−1) was significantly higher (p = 0.03) than in the fore crest zone (1.1 ± 2.6 cm year−1) during the first period. Also, in this crest the rate of increase in the width (back: 2.6 ± 1.8 cm year−1, fore: −0.4 ± 2.6 cm year−1, p = 0.0002) and height (back: 2.6 ± 1.8 cm year−1, fore: −1.1 ± 2.6 cm year−1, p = 0.001) of the fragments was greater in the back crest zone with respect to the fore crest zone for the second period. However, the growth tended to be higher (p > 0.05) in the fore crest zone for some monitoring periods, except in Playa Baracoa (p < 0.006) and El Peruano (p < 0.02) crests during the first and second periods respectively, where the differences in growth were significant (Fig. 4, Table S3). Additionally, the interaction fore crest zone in Rincón de Guanabo (estimate = 8.6; p = 0.02) and El Peruano (estimate = 5.3; p = 0.04) crests had a positive effect on growth rates of the fragments.

Figure 4 Growth rate of the fragments located on the fore (light gray) and back (dark gray) zone of the crests at Playa Baracoa (PB), Rincón de Guanabo (RG), El Peruano (Pr), and Mariflores (Mf) during (A) the first, (B) second, and (C) third study periods.

The numbers above the bars indicate the total number of live fragments or those that were found during each period. The asterisks indicate significant differences between growth in the back and fore crest zones.

The dissolution of the plaster discs positively influenced (estimate = 0.03; p = 0.008) the growth rates, with values above 68% associated to increased growth of the fragments (Fig. 5). Temperature promoted growth (estimate = 0.002; p < 0.001) during the second period. However, the interaction between dissolution and temperature was negative (estimate = −0.001; p = 0.007). The depth effect on growth depends on the crest site and zone. The interactions depth and El Peruano crest was positive (estimate = 8.6; p = 0.002), while the interaction between depth and the fore zones was negative (estimate = −0.009; p = 0.04).

Figure 5 Effect of dissolution of the plaster discs on the growth rates of Acropora palmata fragments.

Temperature

The mean temperature increased from 26.1 ± 0.4 °C in January to 28.1 ± 0.7 °C in May during the first period, in Playa Baracoa and Rincón de Guanabo crests. During the second period (June to September 2022) the mean temperatures were high at 29 and 30 °C. The maximum recorded temperature was 33.6 °C in June. The third period extended from October 2022 to March 2023, when the temperatures decreased, with a minimum temperature recorded of 24 °C (Figs. 6A and 6B). In Jardines de la Reina crests, the temperature was only recorded in July 2022 during the first period, when the mean temperature was 30.6 ± 0.4 °C. During the second period (August to November 2022) the minimum temperature recorded was 28.6 °C in November and the maximum was 31.7 °C in September. During the third period (December 2022 to April 2023) the temperatures decreased till March. The minimum was 26 °C in February and March and the maximum was 32 °C in December (Fig. 6C). The average temperature was approximately 1 °C higher in Jardines de la Reina crests relative to the northwest crests.

Figure 6 Mean seawater temperature (°C); mean ± Standard deviations (SD), in (A). Playa Baracoa, (B). Rincón de Guanabo and (C). Jardines de la Reina reefs.

The blue bars represent the months corresponding to each study period. The numbers at the top indicate the number of days included in each monitoring period.

Discussion

Coral restoration is a practice that requires stronger integration between research and field practices. Effectiveness or success in coral restoration has been linked to two indicators: survival and growth of the coral fragments. Incorporating environmental factors and ecological processes into restoration planning is increasingly recognized as a fundamental component of successful restoration strategies for maximizing the outcomes (Hein et al., 2017; Ladd et al., 2018). Our results suggest that the survival and growth of A. palmata fragments are influenced by specific environmental factors, including water flow, temperature, crest zone, crest, and local site conditions. These findings underscore the importance of incorporating small-scale environmental variability into restoration planning strategies.

The survival of the A. palmata fragments was high (Sp > 0.6) among the four crests. However, our findings indicate that fragment survival was significantly influenced by crest location, with the highest values at Playa Baracoa and El Peruano. The outcome at Playa Baracoa crest was unexpected given its exposure to multiple anthropogenic stressors and low A. palmata density (0.2 ± 0.05 colonies m−2; Ramos et al., 2024), conditions that would normally be considered unfavorable for fragment establishment. This crest receives untreated wastewater from the Latin American School of Medicine. Although the specific nature and intensity of contamination at the site are not well characterized, anecdotal evidence and personal observations report that chemical odors from the school occasionally reach the crest (Ramos et al., 2024). In addition, available water quality data indicate that reefs near Playa Baracoa are more contaminated than those near Rincón de Guanabo (Rey-Villiers et al., 2020; Rey-Villiers, Sánchez & González-Díaz, 2021).

Nonetheless, Playa Baracoa crest harbors a relatively high density of the herbivorous sea urchin D. antillarum (19.3 ± 14.4 ind 10 m−2; González-Díaz & Suarez, 2024), which may contribute to algal regulation; and decrease algae-coral competition (McCook, Jompa & Diaz-Pulido, 2001). Although macroalgal cover is high (>75%), the assemblage includes not only fleshy macroalgae but also crustose coralline algae and other morpho-functional groups that play distinct ecological roles (Ramos et al., 2024). In addition, the fragments were collected from colonies within the same crest where they were transplanted, to ensure local adaptation to the environmental conditions of the site.

On the other hand, El Peruano crest has more favorable conditions than the crests of the northwestern region. This crest remains well conserved with minimal human impact, and its nutrient input mainly comes from nearby lagoons, mangroves, sediments, and oceanic waters (Rey-Villiers, Sánchez & González-Díaz, 2021; Figueredo-Martín, López-Castañeda & Pina-Amargós, 2023). In this study, the higher survival observed in A. palmata fragments at El Peruano, compared to Mariflores, may be related to differences in environmental conditions between these two crests. El Peruano crest lies to the west from JRNP, while Mariflores is located toward the east. According to Hernández-Fernández & Bustamante López (2019), the western part of JRNP has the highest abundance of A. palmata, probably due to topographic differences between the west and east and the influence of regional current systems. The eastern part is closer to the mainland, and receives greater inputs of organic matter, nutrients and sediments from terrestrial sources. In contrast, the western part is more than twice as far from the mainland and may experience reduced terrestrial influence. These environmental contrasts may help explain the better survival outcomes in El Peruano crest.

Fragments placed in the fore crest zone exhibited higher survival than those in the back crest zone. However, the statistical model did not detect a significant effect of the dissolution of the plaster discs (proxy for water flow) on fragment survival between zones. Notably, the interaction between crest and zone was significant at Rincón de Guanabo, where survival in the fore crest zone was substantially greater than in the back crest zone, indicating that localized environmental conditions within each crest modulate fragment survival. The negative influence on survival of the fragments in the back crest zone in Rincón de Guanabo was possibly due to the high Dictyota cover (field observations). This macroalgae was removed at the beginning of the experiment but rapidly overgrew the fragments. Van Woesik, Ripple & Miller (2018) reported that a cover greater than 15% of Dictyota affects survival and growth of A. cervicornis, or habitat conditions that support Dictyota are not conducive to Acropora survival. This indicates that microhabitat characteristics at outplanting sites are critical determinants of fragment survival.

Overall, the growth rates recorded in this study were slower than those reported for wild A. palmata colonies, which typically range from 5 to 10 cm year −1 (Gladfelter, Monahan & Gladfelter, 1978). The slow growth of outplanted fragments was similar to the rates recorded in storm-generated fragments, reported at 1.8 cm year −1 (Lirman, 2000). The stress caused by hurricanes could be like the trauma induced by the collection and transplanting process, also known as initial transplantation shock (Hughes, 1984; Forrester et al., 2012; Forrester et al., 2014). In addition, fragment performance might be influenced by factors such as genotype, identity of the symbiont, fragment physiology, abiotic characteristics of the crest and temperature (Lirman et al., 2014; Papke et al., 2021). Negative growth rate recorded during some periods of the experiment is attributed to tissue loss caused by corallivory or bites by herbivorous fish.

The dissolution of the plaster discs did not indicate a marked gradient of water flow between the crest zones, during April and June, but it was significantly higher in the fore crest zone in December at El Peruano and Mariflores crests, indicating temporal variations during the monitoring periods. The statistical model demonstrated that water flow plays a significant role in fragment growth. Dissolution exerted a significant, threshold-dependent influence on growth: below approximately 68% dissolution, growth declined slightly, whereas above this threshold, growth increased with further water flow. This response suggests that moderate water flow may limit the fragment growth. However, once a critical level of hydrodynamic energy is surpassed, flow probably enhances nutrient delivery, gas exchange, increases nitrogen availability, enhances particle capture efficiency, sediment clearance, facilitates waste product removal, improves metabolism (photosynthesis, respiration) thereby promoting growth (Dennison & Barnes, 1988; Sebens & Johnson, 1991; Lesser et al., 1994; Grigg, 1998; Sebens et al., 2003).

The growth of Acropora spp. has been linked to the rate of change in sea surface temperature (Crabbe, 2007), with the highest growth rates recorded when temperatures range between 28 °C and 30 °C during the warmer months (Shinn, 1966; Bak, Nieuwland & Meesters, 2009). Similarly, our results showed a positive effect of temperature on fragment growth during the second monitoring period, when the highest average temperatures were recorded, 30 °C in the northwestern crests and 31 °C in those of Jardines de la Reina. However, this positive temperature effect was attenuated at higher levels of dissolution of the plaster discs, suggesting that under strong water flow, heat may be dissipated, and water movement probably becomes the primary driver of fragment growth (Nakamura & Van Woesik, 2001; Finelli et al., 2006).

The interaction between crest and zone was significant at Rincón de Guanabo and El Peruano, where growth rates in the fore crest zone was greater than in the back crest zone, indicating that localized environmental conditions within each crest modulate fragment growth. In Rincón de Guanabo, this may be explained by the high cover of Dictyota in the back crest zone, as noted previously, which can inhibit coral growth through space competition and shading (Box & Mumby, 2007). The significant interaction between depth and the El Peruano crest, as well as between depth and the fore crest zones on growth, suggests that local environmental conditions modulated by depth may play a key role in fragment growth. Although, depth alone did not influence fragment growth, it probably affects key environmental variables such as light availability, water movement, and sediment accumulation, all of which are known to affect coral physiology and growth (Baker & Weber, 1975; Pratchett et al., 2015). Also, the growth rate can depend on water currents loaded with higher concentrations of nutrients, resulting from internal waves that are differentially distributed due to the bathymetry of the bottom (Leichter, Stewart & Miller, 2003; Leichter, Deane & Stokes, 2005). Nevertheless, further studies incorporating fine-scale measurements of these factors would be necessary to better understand their role in driving spatial variability in fragment performance.

Conclusions

Our study highlights the importance of incorporating fine-scale environmental variability into coral restoration planning. While overall survival of A. palmata fragments was high across sites, both survival and growth were significantly influenced by site-specific and microhabitat conditions. Growth responses further revealed complex interactions between temperature, water flow, and habitat. These findings underscore the need to move beyond broad site-level criteria and to systematically evaluate microhabitat conditions when selecting restoration sites. Future efforts should integrate ecological knowledge with in situ environmental measurements to optimize coral fragment survival and growth, and to improve the long-term success of restoration interventions under variable and changing reef conditions.

Supplemental Information

Supplemental Information 1 Raw data

Supplemental Information 2 Scripts that were used in the analysis of the raw data

Supplemental Information 3 Initial mean size (±Standard deviation (SD)) of A. palmata fragments in the fore and back zones on the crests at Playa Baracoa (PB), Rincón de Guanabo (RG), El Peruano (Pr) and Mariflores (Mr)

The p-value ≤ 0.05 indicates significant differences between crest zones.

Supplemental Information 4 Significance value (p ≤ 0.05) of non-parametric Kruskal–Wallis (K-W) and Dunn tests for initial sizes (width and height) of A. palmata fragments collected at Playa Baracoa (PB), Rincón de Guanabo (RG) El Peruano (Pr) and Mariflores (Mf) crests from

The p-value ≤ 0.05 indicates significant differences between crests.

Supplemental Information 5 The growth rate (cm year−1 in width and height) (±standard deviation, SD) of Acropora palmata fragments located in the fore and back crest zones for each study period in Playa Baracoa (PB), Rincon de Guanabo (RG), El Peruano (Pr) and Mariflor

The p value ≤ 0.05 (t-Student) indicates significant differences between zones, as indicated in hold. N-value represents the number of total fragments measured for each time. When the value is negative, there was a decrease in growth rate.

We thank Giuseppe Omegna and the crew of OFY, especially the diver’s assistant Noel Lopez Fernández and Maydel Pérez Valle, Anthony Sardiñas, Fabian Pinas Amargos, Tamara Figueredo Martín and others who made this study possible.

Additional Information and Declarations

Competing Interests

Author Contributions

Field Study Permissions

Data Availability

Anastazia T. Banaszak is a Section Editor for PeerJ.

Amanda Ramos Romero conceived and designed the experiments, performed the experiments, analyzed the data, prepared figures and/or tables, authored or reviewed drafts of the article, and approved the final draft.

Patricia González-Díaz conceived and designed the experiments, performed the experiments, authored or reviewed drafts of the article, and approved the final draft.

Gabriela Aguilera Pérez performed the experiments, authored or reviewed drafts of the article, and approved the final draft.

Anastazia T. Banaszak conceived and designed the experiments, authored or reviewed drafts of the article, and approved the final draft.

The following information was supplied relating to field study approvals (i.e., approving body and any reference numbers):

Field experiments were approved by the Ministerio de Ciencia, Tecnología y Medio Ambiente.

The following information was supplied regarding data availability:

The raw data are available in the Supplementary File.

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
