# Peer review of "Does reef crest zone selection influence Acropora palmata (Lamarck, 1816) fragment survival and growth?"

_PeerJ, doi:10.7717/peerj.20303_

## Round 0.1 · original submission · Major Revisions

· Academic Editor

Major Revisions

The reviewers each found useful and interesting results in your work, but also identified sone inconsistencies, problems, and places where additional work could greatly improve its value and impact. I summarize some of the main points and add some suggestions as well, here.

To begin with, there is some confusing terminology. You identify "back crest zones" and "fore crest zones", but never explicitly define or explain how you determined these locations in the different reef areas. This is important for a reader to understand. Next, you have four reef locations (Play Baracoa, Ricon de Guanabo, Mariflores and El Peruano). These appear to be described at various points in the manuscripts and tables as "crests" (e.g. in the abstract) and at others as "reefs" (e.g. in Table 3); this is confusing...please adopt a consistent terminology.


Methods:
Experimental design is relatively clear, although as the reviewers note there are inconsistencies in sample sizes and deployment periods for plaster disks. These need to be checked and corrected.

Important, as the reviewers also note, methodology for determining growth rates, and especially for determining whether the coral fragments are living or dead are unclear or entirely missing.

Statistical analyses need some attention. t-tests have been use to compare various data between back and fore crests, but there are four different reefs to consider. Treating these as independent leads to a weak analysis. A more sophisticated and powerful analysis that would support generalizations, would involve an analysis of variance using 'reef' as a factor and exploring the possibility of interactions. The "model" analysis might be getting at this, but few details are given of how this was done...a regression model may not be the best approach. Reviewer 3 has other statistical suggestions that you should consider.


Results:
Several aspects of the analysis are problematic.

Generalization are made about the differences in water flow based on the plaster disk data. Two of the reviewers accept the conclusion that there were no differences in water flow and call the whole premise of the study into question. But the reality appears to be more subtle. Table 1 actually suggest that there were differences...but that they depended on time. In December, it certainly looks like water flow was higher in the fore reef than the back reef...but this was certainly not true in April...if anything the contrary. water flow can certainly change with season and this needs to be considered. This suggests that the analysis needs to be broken up by season, and it looks like this could be done. Table 1 is hard to interpret and would be much clearer if presented as a Figure.

Calculated survival probabilities are given (Table 2), but there is no survival data. We need to understand how survival was determined and see the data...not just the modeling. The Table is a poor presentation...a Figure would be much superior and should include the data and the modelled fits (see reviewer 3 comments).
Growth rate data are one of the most valuable elements in the manuscript, but Table 3 is a weak presentation; a Figure (two panels, one for with and one for height) with data organized in time would be clearer and much more valuable.
All three reviewers are confused by the presentation of temperature data without analysis. This is a great data set and should be used in context with growth and survival data. There is the possibility to break the data into discrete segments of time. This is something that needs to be pursued in a revision.


Discussion:
At present, this is weak and non-specific. Reviewer 3 has provided several ideas for improvement, and placing results in a clear context with other published work.

Reviewer 1 ·

Basic reporting

1. Some instances of incomplete sentences: lines 233-236, lines 257-259, lines 277-280

2. Some ambiguous/inconsistent language used: phrases like "on the other hand", line 209: ambiguous- write out the names of the two sites located on the northwest crests, line 229: rapidly overgrowth should be rapidly overgrew, lines 267-268: "results could be influenced by different biotic and abiotic factors (such as?), etc (see comments in pdf)

3. Results may be organized to better represent data (e.g., discuss all data pertaining to width growth together and then discuss height)

Experimental design

1. The concept of the study is to identify if differences in wave energy between the fore reef crest and back reef crest influence the growth and survival of A. palmata fragments. However, wave energy was not significantly different between fore and back reef crests except at one location. The study should have been conducted at locations where known differences in wave energy exist.

2. Sites were supposed to be different based on distance to shore, coral abundance, diversity of reef species, anthropogenic stressors, geomorphological characteristics, and management and protection. However, such differences between sites were never mentioned or explained, and should be discussed if they are believed to have had any influence on growth and survival.

3. It is mentioned in the Discussion that the stress of collection, transport, and acclimation may have influenced the growth and survival of the outplants, but such methods are not mentioned in the Materials and Methods section for discussion.

4. Differences in temperature are discussed well with the Materials and Methods, but are not considered as a potential source of variance on growth and survival. As mentioned in the Discussion, why wasn't the temperature data suitable for statistical analysis?

5. Methods used to measure growth are not provided.

Validity of the findings

1. Lines 277-280: "linking the water flow gradient induced by the waves along the reef with the survival and growth of the A. palmata fragments is key." However, a flow gradient was not observed at the sites except for one location. Therefore, this study does not find a link between water flow and growth and survival. It is recognized that research is needed to determine how water flow may impact growth and survival of outplants, but this study does not adequately address this.

2. It is mentioned that sites differed in distance to shore, coral abundance, diversity of reef species, anthropogenic stressors, geomorphological characteristics, degradation states, and management and protection. However, there isn't any documentation showing such differences between sites, which prevents appropriate discussion of how outplant success may be based on such conditions.

3. A larger discussion is needed to address the role of wave action within reef microhabitats and how wave energy may act synergistically with other conditions within each microhabitat, especially in light of the lack of difference in wave energy between sites.

Additional comments

See notes and comments within the PDF

Annotated reviews are not available for download in order to protect the identity of reviewers who chose to remain anonymous.

Reviewer 2 ·

Basic reporting

-

Experimental design

-

Validity of the findings

The work is an important effort to contribute to knowledge about the factors that can affect the effectiveness of restoration processes. In this case, the areas described as having high wave energy are those where Acropora palmata has historically lived. However, in the last two years, 2023 and 2024, the El Niño phenomenon increased temperatures, making it impossible for the species to live in shallow areas on many Caribbean reefs. The document does not mention the depth of the sampling sites, and in that case, how could the work recommend reef crests when, due to global warming, they might not be effective for restoration?

The abstract reports a different number of samples than described in the methods section.
The median dissolution data were used for nonparametric statistical analyses; however, it is not clear why they express the percentage of dissolution relative to the median.
From lines 170 to 173, the references to TS1 and TS2 appear to be reversed.
If approximately four visits were made to each zone, why were dissolution plates not performed on each sampling date?
The survival rate for line 177 and the growth rate for line 187 are from the same fragments, at the same sampling times per reef zone. Therefore, it is unclear why Table 2 shows survival rates in days for each zone, while Table 3 presents fragment measurement data up to 400 days.
Lines 204 to 207 present regressions, but these results are not seen in the document.
Lines 209 to 221 describe the temperature during the study, but do not provide any information regarding the growth or survival of the fragments.
Lines 240 to 241 describe what growth rates Lirman obtained.
Lines 241 to 250 present a list of possible factors that may affect growth. This is good information for the introduction, but it lacks a comparison with the results obtained, since none of these factors were measured.
Lines 250 to 253 address temperature and provide a discussion, but then mention that no analysis was performed, so the temperature is not relevant.
Lines 269 to 270 previously mentioned that the fragments came from the same areas, so acclimatization to a restoration site would not apply to this discussion.
Lines 272 to 282 are well oriented to the results of the work, but not 282 and 283, as they do not have information on the condition of the microhabitat.
There is a considerable amount of literature on annual growth rates for Acropora palmata. The document is limited to daily increments. However, if you review this literature, you can contribute to the annual growth rates of the species and improve the discussion.

Annotated reviews are not available for download in order to protect the identity of reviewers who chose to remain anonymous.

·

Basic reporting

The manuscript looks to assess the role of water flow in the survival and growth of Acropora palmata colonies outplanted to fore reef and back reef zones at four reefs around Cuba. The authors used plaster dissolution as a proxy for hydrodynamic energy and measured survival, coral width, and coral height as response variables to assess the contribution of location within a reef and between reefs of the region. A small to negligible difference is plaster dissolution taken over 24- 48 hours during the ~430 days of outplanting, suggesting wave energy was similar across zones and reefs. Survival was high across outplanting sites, apart from reduced survival at the back reef zone at one site, potentially tied to macroalgae growth. Coral growth was more variable between sites and zones than survival, but low growth rates meant the significant difference between size and zones was small. Taken together, the authors conclude that wave energy is unlikely to be a determining factor in A. palmata growth and survival at these reefs, pointing to other factors as the predominant variables creating the observed trends in the current study.

The study offers a useful insight into reefs that are/will be used for restoration, evaluating survival probability and growth prior to massive restoration will be important for strategizing. The site design encompasses a large spatial area around Cuba, which offers a chance to evaluate broad differences in the fore vs back reef zones. While the experimental design is sound, I feel there are important aspects to address in the analysis and interpretation, and therefore the presentation, of the study’s findings. In particular, the main conclusion that variation in coral performance was not explained by variation in wave energy, is somewhat incomplete because there were little to no variation in wave energy as well as little to no variation in performance (no effect in survival of crest/zone, and significant but minor effects for growth). One alternative interpretation of this could be that wave energy does indeed impact survival, but because wave energy was not different between zones, performance was not different between zones. I believe the authors somewhat addressed this by shifting focus to reef/zone comparisons, but ultimately, the model design could be reviewed to better address this approach (suggestions below). The eventual focus on microhabitat variation explaining the trends seen is possible, but I feel the statistical and logical arguments need to be better delivered to support this claim. Finally, a deeper discussion that addresses some of the key limitations of the study would allow a broader use of supporting literature while still addressing the hypotheses of the manuscript.

In terms of the delivery of the material, there were frequent omissions of important details and incongruencies throughout the text that impact the interpretation of the manuscript. I have outlined them below, but I recommend careful review of the material for coherence between aspects of the text.

Line 111 states plaster discs were left for 24 hours at Playa Baracoa and Rincón de Guanabo reefs, but Table 1 shows that they were placed for 48 hours.

The abstract states there were 60 fragments in total, and line 119 says 50 fragments were used.

The text describes the start dates for the experiments to be either January or February 2022 and mentions that temperature loggers were recording throughout the experimental monitoring period. However, Figure 4 shows that only one set of temperature logs started in January 2022 (Rincon de Guanabo) while the others started in April or July. It is unclear which start date is correct, but if the latter is correct, then it would be hard to compare coral survival/growth across vastly different parts of the year. Also, there is only data from 3 loggers, but the text does not mention that one logger was used to measure the temperature at Jardines de la Reina reefs. This should be mentioned in the methods section at least, and potentially discussed for why 1 logger is likely to be representative of the two sites.
Line 145: What is survival probability? How was it calculated?

Line 124: Need to describe how heights and widths were measured. What part of the coral were they measuring from? The largest plate? What if there were multiple plates?

Line 166: Is the percentage dissolution an average of the four discs or the sum total? If the former, then there needs to be an error or range associated with it.

Line 173: Diameter is mentioned for the first time, but it is unclear if diameter is synonymous with width? I recommend keeping the same term throughout

Line 209: a similar pattern to what?

Line 234: The sentence begins with despite but does not offer a contrasting point. Perhaps this was an incomplete thought? The rest of the paragraph is dependent on this statement.

Line 258-259: Seems like another incomplete thought

Table 2: Should include the additional metrics of the results of the models comparing survival

Table 3 would be better displayed as a plot to visualize the change in growth rates over time and space. This presentation makes it nearly impossible for readers to easily compare the different permutations of reefs*zones.

Experimental design

Line 101-104 states that “These reefs differ in distance from the coast, abundance, and diversity of reef species such as fish and sea urchins, anthropogenic stressors, geomorphological characteristics, and management and protection,” however there is no discussion in the text of how these parameters changes across the study reefs. This is briefly referenced when mentioning Dictyota overgrowth as a potential cause for reduced growth/survival at Rincon de Guanabo, but this was only anecdotal. It sounds as though these biotic and abiotic differences were important to mention but never explored or discussed – perhaps these could be incorporated into the models for survival and growth to be better predictors compared to site and zone. Given the ultimate conclusion of microhabitat variation, a greater effort to describe the microhabitats should be employed. This may be as simple as a supplementary table summarizing findings of these reefs from the literature. If the previous literature does not explore these exact sites as I interpreted from the text, there should be some way to infer why, for example, RG has more nutrients or distance to the mainland than Mariflores (this is just a hypothetical here).

The timing of plaster disc measurements and coral outplanting is of particular concern for addressing the key hypotheses of the study. Specifically, the explanation for the hypothesized increased survival/growth in the fore reef zone was increased wave energy relative to the back reef zone. However, the wave energy was only measured one to two times over the year+ experimental period, making the resulting dissolution rates likely uncharacteristic of the conditions experienced by the experimental coral over the entire outplant. It may have been logistically impossible to deploy the plaster discs multiple times, however, this constraint should at least be included in the discussion, as it is vital to the interpretation of the results. The authors should reconsider using wave energy as the defining theme of the manuscript unless the 24-48 hr dissolution rate measure can be shown to be representative of the average wave energy of the zone. Otherwise, readjusting the focus to site/zone differences would allow the hypothesis to be better addressed within the manuscript. As it stands, the resolution of the wave energy measurements does not seem strong enough to support or refute the main hypothesis. Sourcing published data on the geomorphological characteristics that are mentioned to be recorded at these reefs (line 103) could allow some inference of the general differences between general wave energy between zones, given the limited in situ data collected here.

Modelling Improvements
A model predicting variation in survival and growth to include the wear dissolution variable, given that the main hypothesis focused on wave energy’s effect on coral performance. This could be easily incorporated and would better address the link between wave energy and growth/survival but allowing a correlation between wave energy and coral performance, rather than using two separate models to assess wave energy vs site/zone, then site/zone vs performance. The aspect of the hypothesis focusing on the difference between fore and back zones would be addressed by evaluating zone as a nested variable within crest, given the experimental design here. This would allow the difference between zones to be compared within a site, which may be more beneficial because the sites likely differ more between each other than the zones do, environmentally speaking (evidenced by Table 1). The temperature in each period that which growth was measured could also be incorporated for a direct comparison of the temperature’s effect on growth rate. (Line 254 suggests this is not possible, but given what is described in the methods for temperature data collection, it should be.) Finally, if the authors were hoping to identify factors influencing the results observed, it may be useful to generate more environmental data for the reefs they evaluated. This may be possible given that a variety of abiotic and biotic differences were cited in lines 101-104, as well as with temperature. Incorporation of multiple environmental parameters can be done in a Bayesian approach, as in https://www.frontiersin.org/journals/marine-science/articles/10.3389/fmars.2020.00163/full or with a PCA that shows how the reefs are different and which characteristics are driving the differences. I believe these approaches would more directly address the hypotheses outlined in the introduction.

Finally, there are no full model results for the survival and growth-based models. For example, the growth rate included a crest*zone interaction, but it is unclear if that was significant or not? Table 3 includes a p-value, but it is unclear what test this is from, and the estimates of effect sizes are also not included. This makes it hard to identify what comparisons were included in the models.

Interpretation of temperature
I believe there is a missed opportunity to utilize the high-resolution temperature data collected on the reefs. Specifically, there is only one line in the results related to between reef temperatures. However, a more in-depth analysis may reveal explanatory variables for the differences between reefs. While inclusion of this is not crucial, it would better support the advanced statistics described above and make better use of the high-resolution temperature data collected.

Validity of the findings

Overall, the reporting of the results of the models was thorough, and I appreciated the amount of work that was required to address that level of detail. The results based on the tests completed seem sufficient however, the interpretation and the discussion of these results within the context of the experiment and the existing literature seem tenuous. In particular, the discussion of findings seems to state a result but supply a justification or potential explanation that does not have a clear line from the data or the existing literature – provide examples under “improved depth of discussion”.

Influence of the zone on growth
The authors take great care to outline the differences in growth for both height and width between the zones of the four reefs within the results section. Interestingly, growth in width seems to often be significantly different between fore and back zones, while height showed the opposite trend for many sites. Yet, the discussion uses the non-significant linear model of zone on growth to claim there is no effect of zone on growth. While the growth rates are very low (with standard deviation larger than the difference between zones in many cases), I wonder if the nested approach described above will allow the variation between sites to be ignored to compare between zones within a site. The discussion seems quick to dismiss the role of zones despite significant (though small) differences between the zones on a site-by-site basis. Given the introduction discussed the potential for adaptive morphological changes (Line 76-78), it may be possible that the high survival overall, regardless of zone (and potential accompanying wave energy), is a result of variation in growth form (height vs width)?

Improved depth of discussion
There are various points made in the discussion that do not appear to be supported by the text or have weak links with the cited literature. For example, Line 239 offers local adaptation as an explanation for, presumably, the high survival across sites. This may be true, but without a transplant experiment, it is hard to really infer local adaptation. While possible, alternative theories could be included here so it does not sound like the authors have only considered one possible explanation. Similarly, Line 245-247 states “The growth rate varied between the reef zones, depending on the study period, possibly due to water currents loaded with higher concentrations of nutrients, resulting from internal waves that are differentially distributed due to the bathymetry of the bottom” and offers a citation to support. This direct causal relationship does not seem supported by the data, given that no nutrient data were collected for the study. While the citation supports a possible mechanism creating the environmental conditions, the language used here seems to assign a direct, causal relationship that cannot be supported here.

The exploration of the interesting result that the back reef corals at El Peruano were growing more than in the fore reef, AND that the flow was significantly lower in the back reef, was quite limited. The subsequent discussion of a single study on the flow effects on P. damicornis states increased flow is favored (this is a somewhat ambiguous term) by photosynthesis and respiration then mentions the growth rate of P. damicornis can be similar in sites with different water flow. This seems to be used as an argument for the growth of A. palmata being influenced by factors other than flow, yet there is no discussion of these different factors from the background literature or the current data set. This discussion seems to dismiss a potentially interesting difference and instead assigns a combination of unexplained variables to explain the reported results. This is insufficient for the level of nuance provided in the results and given the history of work exploring flow on coral reefs. I struggle to find the relevance of Lesser et al. 1994 in this paragraph as written. It feels as though Lesser is being used as an example of flow not influencing growth rate in P. damicornis, and this is being used as the basis for the interpretation of growth rate being controlled by other factors. I suggest incorporating more studies to provide context for other environmental factors influencing coral growth, rather than using 1 study to contextualize the lack of relationship between wave energy and growth.

The potential stress of “collecting, transporting, and the possible cost of acclimation of coral fragments to the restoration site could be a factor” (Line 269-270) seems inappropriate as it is used to explain the variation in growth rates between sites given that, presumably, all corals went through the same handling process? If this is not the case, then the methods should describe any differences. If this is being used to provide an explanation for the low growth rates or variation in growth between time periods, it should explicitly say that in the sentence. As it stands, it is too ambiguous.

There seems to be an unsupported discussion of microhabitat variation. The final line of the conclusion states that “the transplant success of the fragments in this experiment and the reefs appears to be influenced primarily by microhabitat conditions,” as well as lines 236-237, “This probably indicates that the fragments may be more influenced by the microhabitat characteristics of the outplanting sites”. These statements come after a discussion of similar survival rates across crests, which seem to oppose the effect of microhabitat because there was no variation in survival. The argument for microhabitats influencing growth is stronger, but the justification for this microhabitat variation is limited. For example, the only characteristics describing differences between zones are the dissolution of plaster, but that was dismissed, and temperature was not evaluated between zones. It is unclear if there is a crest*zone interaction in growth, which would provide statistical support that growth was modulated by the particular microhabitat a colony was located in. I believe this could be better addressed with the outlined suggestions for modeling. However, whether or not this line of theory holds up will be dependent on that.

---

## Round 0.2 · Minor Revisions

· Academic Editor

Minor Revisions

Your submission was re-reviewed and the reviewer was very positive about the changes you have made. I concur. They have suggested some edits for clarity, but in particular, I would like you to consider: a) a clearer explanation of the origin of the coral fragments (comments on line 162), b) further clarification and caveats on the estimates of survival probabilities (comments related to line 200), and c) some comment about the apparently higher growth rate in the fore crest zone (comments related to line 257).

I think these can be accomplished fairly easily.

·

Basic reporting

The authors did a good job incorporated the reviewer suggestions during revision and this is particularly evidenced by the greater discussion of the nuance seen within the results. With this additional context, the scope of the manuscript becomes clearer and better displays the concept the authors seem to want to convey – that a combination of local factors influence restoration success and zone within the same reef can be crucial to consider for restoration. The remaining comments I have would help make aspects of the study clearer for readers but incorporating these changes would not significantly affect the manuscript content.

Basic Reporting: The reporting within the manuscript has drastically improved with the authors revisions. Minor comments for clarity are included below.

Experimental design

Experimental Design: The authors incorporated changes to the manuscript that better leverage the data they have and account for weakens in experimental design that cannot be corrected. Improved modeling better accounts for the nuance inherent in the experiment. Model interpretation within the manuscript can be improved with the comments made below.

Validity of the findings

Validity of the findings: By including more nuance in the data analysis and results, the authors created a stronger set of findings. Including information on the origin of coral fragments as discussed below will be important for replicating this study and providing study-specific context for the findings. Raw data is included but if additional transparency is desired then analysis scripts could also be included here or made available in a github repository.

Additional comments

Abstract:
The line “The growth rates were slower ( -1.5 to 7.3 cm year -1 ) than those reported for wild A. palmata colonies and were negatively affected (estimate = -6.1; p = 0.004) in the fore crest zone.” Make it sound like wild colonies were also measured in this study which from what I understand were not assessed. I would add “previously recorded” just to show the comparison you are making is with other studies and not necessarily those from Cuba or these reefs.
Introduction:
Line 103: add citation for the statement that A palmata morphology, growth, and branching is affected by waves and water movement
Methods:
Line 132: Include the names of the crests rather than just the national park since Figure 1 does not reference the park but the crest names in that region
Lin 162: There is no discussion of the origin of the coral fragments however in the discussion it is mentioned that colonies originated from the reefs at which they were outplanted to rather than a common garden. This is vital information and should be mentioned in the methods. Moreover, if possible the authors should include which zone the colonies came from (fore vs. back) as this will be important to understanding local adaptation. Moreover, if possible genet information would be good to include if tracked and replicated within a site.
Line 200: I understand the reviewer’s confusion about the survival probability. While I do not think the authors need to describe the Kaplan-Meier method in full, it would be good to note that this method incorporates time of death rather than presence to estimate survival probability – therefore later death is weighted less than early death (or something along those lines). I think this is important because readers, may typically look for the hazard ratio associated with site or zone factors (as you do to compare individual levels of the factors within the methods) while the final survival probability looks very similar to a proportion of surviving individuals metrics. The authors should also include a reference for the R packaged used for the survival analysis
Line 200 + 216: Pairwise comparisons of levels within factors for the survival and lme4 analyses typically present estimates relative to a reference level for categorical factors. For example, El Peruano has an estimate of 12.9 for the survival analysis which is in reference to some other crests. Therefore, it is important to include what the reference level is for estimating effect sizes because estimates are compared to only the reference and not necessarily every other level as they are in tests like TukeyHSD. Typically the default is the first level of the factor (usually alphabetical if you don’t reset the levels). These reference levels should be put in the methods.

Results:
Line 257: the fact that growth was higher in fore crest zone in other time periods is jarring because it is only briefly mentioned but is in contrast to the lengthy results reporting showing the back crest zone had higher growth in the third time period. Upon review of Table S3, it looks like these comparisons in the first and second time periods were not significant. Therefore, I would include the p-value (something like p > 0.05) at the end of this sentence to clarify that the growth in fore crest zones only tended to be higher and therefore did not warrant addition exploration.
Figure 4: add significance indicators (like an *) to help with visualization
Line 279: what aspect of temperature was 1C higher in Jardines de la Reina? Average temperature?

---

## Round 0.3 · accepted · Accept

· Academic Editor

Accept

All of the issues raised on the second review have been clearly addressed.